# The Relationship between Character Traits and In Vivo Cerebral Serotonin Transporter Availability in Healthy Subjects: A High-Resolution PET Study with C-11 DASB

**DOI:** 10.3390/ph16050759

**Published:** 2023-05-18

**Authors:** Jeong-Hee Kim, Hang-Keun Kim, Sang-Wha Lee, Young-Don Son, Jong-Hoon Kim

**Affiliations:** 1Biomedical Engineering Research Center, Gachon University, Incheon 21936, Republic of Korea; jhkim1104@gachon.ac.kr (J.-H.K.); dsaint31@gachon.ac.kr (H.-K.K.); 2Department of Biomedical Engineering, College of IT Convergence, Gachon University, Seongnam-si 13120, Republic of Korea; 3Neuroscience Research Institute, Gachon University, Incheon 21565, Republic of Korea; 4Department of Chemical and Biological Engineering, Gachon University, Seongnam-si 13120, Republic of Korea; lswha@gachon.ac.kr; 5Department of Psychiatry, Gachon University College of Medicine, Gil Medical Center, Gachon University, Incheon 21565, Republic of Korea

**Keywords:** character traits, positron emission tomography, serotonin transporter, C-11 DASB, healthy subjects

## Abstract

To elucidate the potential roles of serotonergic activity in human character traits (i.e., self-directedness, cooperativeness, and self-transcendence), we investigated the relationship between these character traits and serotonin transporter (5-HTT) in healthy subjects. Twenty-four participants underwent High-Resolution Research Tomograph–positron emission tomography scans with [^11^C]DASB. To quantify 5-HTT availability, binding potential (BP_ND_) of [^11^C]DASB was obtained using the simplified reference tissue model. The Temperament and Character Inventory was used to assess subjects’ levels of three character traits. There were no significant correlations between the three character traits. Self-directedness was significantly positively correlated with [^11^C]DASB BP_ND_ in the left hippocampus, left middle occipital gyrus, bilateral superior parietal gyrus, left inferior parietal gyrus, left middle temporal gyrus (MTG), and left inferior temporal gyrus (ITG). Cooperativeness was significantly negatively correlated with [^11^C]DASB BP_ND_ in the median raphe nucleus. Self-transcendence was significantly negatively correlated with [^11^C]DASB BP_ND_ in the right MTG and right ITG. Our results show significant correlations between the three character traits and 5-HTT availability in specific brain regions. In particular, self-directedness was significantly positively correlated with 5-HTT availability, suggesting that a goal-oriented, self-confident, and resourceful character may be related to higher serotonergic neurotransmission.

## 1. Introduction

Three character traits from the Temperament and Character Inventory (TCI) suggested by Cloninger et al. [1], “self-directedness (i.e., responsibility, purposeful, resourcefulness, enlightened second nature, and self-acceptance)”, “cooperativeness (i.e., helpfulness, empathy, social acceptance, compassion, and pure-hearted conscience)”, and “self-transcendence (i.e., transpersonal identification, self-forgetful, and spiritual acceptance)”, are each essential for general psychological well-being and positive emotions. Healthy subjects with adequate levels of these character traits, especially high “self-directedness”, generally have positive emotions such as happiness (or subjective well-being), joy, satisfaction with life, and optimism [2,3,4]. On the other hand, patients with psychiatric disorders tend to have lower levels of self-directedness and cooperativeness, and higher levels of self-transcendence, compared with healthy subjects [5]. Given these links between character traits and psychopathology, a better biological understanding of character traits may improve our understanding of the pathology of psychiatric disorders.

Like character traits, the serotonergic system is also associated with general psychological well-being and positive emotions [6,7,8,9,10]. In particular, serotonergic neurotransmission has been linked to several brain functions, such as emotional regulation and cognitive performance [11], and altered serotonergic neurotransmission has been implicated in several psychiatric disorders, including anxiety disorders and depression [12].

Moreover, several genetic studies in humans have shown relationships between human character traits and the serotonergic system [13,14,15,16]. Notably, these studies reported significant associations between self-directedness and various serotonin (5-hydroxytryptamine, 5-HT) genes, including primarily 5-HT transporter (5-HTT) or 5-HT_2A_ receptor (5-HT2AR) genes.

However, few positron emission tomography (PET) studies have investigated the associations between character traits and serotonergic neurotransmission in vivo. Only two PET studies have reported relationships between character traits and 5-HTT availability in healthy subjects [17,18]. One previous study by Kim et al. found that self-transcendence and spiritual acceptance in 16 healthy individuals were significantly negatively correlated with [^11^C]DASB binding in the caudal and median raphe nuclei, respectively [18]. In another previous PET study that assessed brain 5-HTT density in vivo in 21 healthy subjects with high and low levels of harm avoidance, secondary analysis showed that self-directedness was significantly positively correlated with [^11^C]MADAM binding in the dorsal raphe nucleus [17].

Therefore, we investigated the relationship between three character traits and in vivo cerebral 5-HTT availability in healthy individuals using High Resolution Research Tomograph (HRRT)-PET with [^11^C]DASB, a radioligand that specifically binds to 5-HTT, to elucidate the potential roles of serotonergic activity in these character traits.

## 2. Results

The mean raw scores for self-directedness, cooperativeness, and self-transcendence scales were 54.7 ± 7.7, 59.5 ± 7.9, and 22.7 ± 13.3, respectively (Table 1). There were no significant correlations among the three character traits (*r* = −0.09 to 0.22, *p* > 0.05). The raw scores of these character trait scales were not significantly correlated with age (*r* = −0.32 to 0.23, *p* > 0.05). There were no significant differences between males and females in the raw scores of cooperativeness and self-transcendence scales (t = −0.80 to −0.25, *p* > 0.05), but the raw score of self-directedness scale was significantly higher in males than in females (t = 2.43, *p* = 0.024).

Region-of-interest (ROI)-based partial correlation analysis with age and sex as covariates revealed that self-directedness was significantly positively correlated with [^11^C]DASB binding potential (BP_ND_) in the left middle temporal gyrus (MTG) (*r* = 0.428, *p* = 0.047), left inferior temporal gyrus (ITG) (*r* = 0.433, *p* = 0.044), bilateral superior parietal gyrus (SPG) (right: *r* = 0.488, *p* = 0.021, left: *r* = 0.508, *p* = 0.016), left inferior parietal gyrus (IPG) (*r* = 0.593, *p* = 0.004), left middle occipital gyrus (MOG) (*r* = 0.542, *p* = 0.009), and left hippocampus (HIP) (*r* = 0.477, *p* = 0.025). Cooperativeness was significantly negatively correlated with [^11^C]DASB BP_ND_ in the median raphe nucleus (rapheM) (*r* = −0.497, *p* = 0.019). Self-transcendence was significantly negatively correlated with [^11^C]DASB BP_ND_ in the right MTG (*r* = −0.450, *p* = 0.035) and right ITG (*r* = −0.446, *p* = 0.037). Details of these results are shown in Table 2 and Figure 1.

Further voxel-based partial correlation analysis with age and sex as covariates revealed significant positive correlations between self-directedness and [^11^C]DASB BP_ND_ in the temporal pole of the left superior temporal gyrus (STG) (*p* < 0.0001), right calcarine fissure and surrounding cortex (V1) (*p* = 0.0001), left IPG (*p* = 0.0001), left MOG (*p* = 0.0002), and left MTG (*p* = 0.0002), and a significant negative correlation between self-transcendence and [^11^C]DASB BP_ND_ in the right ITG (*p* = 0.0002). Details of these results are shown in Figure 2.

## 3. Discussion

In this study, we found that the 5-HTT availability in specific brain regions was positively correlated with self-directedness and negatively correlated with cooperativeness and self-transcendence, indicating that serotonergic neurotransmission is significantly involved in these character traits in healthy subjects. To date, only two molecular PET studies have demonstrated associations between one or two character traits and in vivo 5-HTT availability [17,18], and to our knowledge, this is the first [^11^C]DASB PET study to report significant associations between three character traits and in vivo 5-HTT availability in healthy subjects.

The main finding of the ROI-based partial correlation analyses was that high self-directedness was associated with high 5-HTT availability in the HIP and some cerebral cortices, such as the temporal, parietal, and occipital lobes. These regions were also consistent with the brain regions observed in the voxel-based analysis, with the exception of the HIP. Neuroanatomical and functional connections between the HIP and several brain regions, including the frontal lobe and structures located in the parietal, temporal, and limbic circuits, are involved in the flexible cognition and social behaviors related to human character traits [19]. Changes in serotonergic activity in these brain areas may lead to modulations of a character trait associated with self-directedness. In an [^11^C]MADAM PET study, higher 5-HTT BP_ND_ in the raphe nucleus was associated with higher self-directedness scores [17]. Neuroimaging studies using functional magnetic resonance imaging (MRI) have shown that neural networks, including the temporal cortex, are involved in appraisal of the self [20]. Gray matter volumes have shown a positive correlation between self-directedness and several cortical regions, including the MOG and STG [21].

The positive correlation between self-directedness and 5-HTT availability in vivo in these brain regions may be associated with genetic factors affecting 5-HTT or with secondary changes in 5-HTT density due to changes in endogenous extracellular serotonin levels. These findings are also in line with previous studies that have demonstrated associations of character traits with neural substrates in the HIP and neocortex [22,23]. 

As shown in Figure 2 and Appendix A, additional automated anatomical labeling (AAL) template region- and voxel-based partial correlation analyses also revealed a significant positive correlation between self-directedness and 5-HTT availability in the right V1, one of the brain regions excluded from the 52 a priori ROIs in this study. The V1 is located in the occipital lobe, and its function is to perceive and process visual information, as well as to be involved in complex visual perceptual processes. Sensory processing, such as visual perception processing, is a natural process of brain functioning by which the nervous system interacts with the environment and has been shown to be associated with human character traits [24,25,26]. A study by Choi et al. reported that positive characteristics were positively correlated with sensory seeking but negatively correlated with low sensory registration, which refers to difficulty noticing or responding to sensory information [25]. Several studies have suggested that sensory sensitive, sensory seeking, sensory avoiding, and low sensory registration are also related to temperament and character traits in adulthood [24,25,26]. A recent study by Kim et al. demonstrated that self-directedness was predicted by low sensory registration and high sensory seeking [27]. These results suggest that specific sensory processing patterns may influence the trait of self-directedness [27]. Based on these reports, serotonergic activity in the occipital lobe, which is involved in visual perception processing, may play an important role in the manifestation of self-directedness.

In this study, cooperativeness was negatively correlated with 5-HTT availability in vivo in the rapheM. The rapheM, one of the rostral raphe nuclei containing the majority of serotonergic neurons in the brain, projects primarily to the amygdala (AMYG), HIP, ventral tegmental area, hypothalamus, and temporal cortex, and modulates emotional and behavioral functions, including mood regulation, cognition, and sleep–wake cycle [28]. Consistent with our results, in a correlational study between half maximal effective concentration (EC50), an index of 5-HT_2_ receptor sensitivity, and personality traits, some subscales of the cooperativeness were negatively correlated [29]. Therefore, the functional activity of the rapheM and its related neural circuitry may be an important biological basis for cooperativeness.

We also found significant inverse correlations between self-transcendence and 5-HTT availability. Self-transcendence has been associated with 5-HTT in a PET study [18] and in genetic studies [30,31,32]. Several genetic association studies reported that self-transcendence was significantly correlated with 5-HT_1A_ receptor (5-HT1AR), 5-HT2AR, and 5-HT_6_ receptor gene polymorphisms [33,34]. Several previous studies have shown that serotonergic hallucinogens cause changes in the 5-HT system, as well as higher-level cognitive and perceptual distortions [35,36]. Furthermore, depressed patients had a significant reduction in self-transcendence scores during treatment with selective serotonin reuptake inhibitors [37]. Based on these reports, the inverse correlation between self-transcendence and 5-HTT availability could be interpreted as high levels of the 5-HT being associated with reduced self-transcendence. Our findings are in line with previous results showing inverse correlations between self-transcendence and 5-HT1AR availability in serotonergic projection areas such as raphe nuclei, HIP, and neocortex [38].

The significant inverse correlation between self-transcendence and 5-HTT availability in ROI-based analysis was found in the right MTG and right ITG. The result in the right ITG in particular was corroborated by supplemental voxel-based analysis. In many clinical studies of temporal lobe epilepsy (TLE), profound spiritual experiences have been reported as a characteristic of TLE [39], suggesting that self-transcendence may be attributable to similar disturbances in the temporal lobe. A previous study by Persinger et al. demonstrated that similar spiritual experiences can be induced by artificially stimulating the temporal lobe [40]. Therefore, in line with the aforementioned studies, our results suggest that 5-HTT availability in the temporal lobe is critically involved in self-transcendence.

We further investigated the relationship between character traits and 5-HTT availability in ROIs for males and females separately using ROI-based correlation analysis with age as a covariate. As shown in Appendix A, males had significant negative correlations between self-transcendence and 5-HTT availability in the right STG, right MTG, and right ITG. Females exhibited significant positive correlations between self-directedness and 5-HTT availability in the medial orbital part of left superior frontal gyrus (SFG), left olfactory cortex (OLF), left SPG, left IPG, left HIP, and right AMYG. Further studies on larger cohorts are required owing to limited sample size (12 males and 12 females) in additional supplementary analyses stratified by sex.

Considering that the harm-avoidance temperament has been suggested to be associated with serotonergic neurotransmission [1,13,41,42,43], we performed an additional supplementary ROI-based analysis with age and sex as covariates. We found a significant negative association between the harm-avoidance temperament and the 5-HTT availability in the left posterior cingulate gyrus (PCG) (*p* = 0.028) (Appendix A), suggesting the association between the harm-avoidance temperament and the pre-synaptic marker of serotonergic neurotransmission.

Our results should be interpreted with some limitations in mind. Metabolite-corrected arterial input-based kinetic modeling methods are ideal for quantification of [^11^C]DASB BP_ND_; however, these methods are invasive as they require radial artery cannulation, and the resulting discomfort might be a confounding factor and would also lead to limited recruitment of participants. Inaccuracies in determining the arterial input function can also be a source of bias in endpoint estimation [44]. Therefore, in the present study, the [^11^C]DASB BP_ND_ values were obtained using simplified reference tissue model 2 (SRTM2) [45] with cerebellar gray matter as the reference region. Reference tissue compartment models, such as SRTM2, use the time–activity curve (TAC) of a reference region with non-existent or very low specific uptake as an input function [46,47]. Although the cerebellum, used as a reference region in the present study, is a region that is not completely free of 5-HTT, a previous postmortem study in the human brain has shown that trace concentrations of 5-HTT are present in the cerebellum and white matter, and that the concentration of 5-HTT in the cerebellum has minimal effect on the quantification of [^11^C]DASB BP_ND_ [48]. Hence, similar to the present study, previous in vivo human brain studies have quantified 5-HTT availability using reference tissue compartment models with the cerebellum as a reference region [49,50,51,52,53]. However, the possibility that the [^11^C]DASB BP_ND_ may be slightly underestimated by the use of the cerebellum as a reference region should not be completely ruled out [48]. Further molecular PET studies using arterial-input-based kinetic modeling methods are required to confirm our results on the relationships between character traits and 5-HTT availability. 

As illustrated in Figure 1, the scores of character traits and [^11^C]DASB BP_ND_ values exhibited substantial variances. The wide dispersion of data points may have resulted in relatively low statistical power in examining the observed correlations. Consequently, our study may have been limited in detecting relatively weak associations between character traits and 5-HTT availability as assessed using the ROI-based correlation analysis.

Future dual-tracer PET imaging studies with radiotracers that bind to presynaptic and postsynaptic markers of the 5-HT system should be conducted to completely understand the complex relationship between character traits and the 5-HT system in humans.

## 4. Materials and Methods

### 4.1. Subjects

This study was approved by the Institutional Review Board of the Gachon University Gil Medical Center, and all study procedures were conducted in accordance with international ethical standards and the Declaration of Helsinki. Written informed consent was obtained from all subjects after a full explanation of the study purpose and procedures before participation in the study. 

Twenty-four healthy subjects (twelve males and twelve females) were recruited through local poster advertisements, and met the following criteria: (i) age between 19 years (legal adult age in the Republic of Korea) and 60 years; (ii) no past or current psychiatric disorders from the Diagnostic and Statistical Manual of Mental Disorders 4th edition (DSM-IV) [54], established using the Structured Clinical Interview for DSM-IV [55]; (iii) no past or current substance dependence/use; (iv) no history of neurological or medical disorders; (v) no past or current use of substances/medications known to affect the central nervous system; and (vi) not pregnant at the date of the PET scan. Their mean age was 29.9 ± 7.9 years (range = 22–44 years, median = 27 years). A board-certified radiologist confirmed that all subjects had no structural abnormalities on their brain MRI. The demographic characteristics of the subjects are shown in Table 1.

### 4.2. Assessment of Character Traits

The TCI was used to measure human character dimensions (i.e., self-directedness, cooperativeness, and self-transcendence). Among these character traits, self-directedness is defined as an individual’s executive ability to regulate and adapt behavior to situations in accordance with personal goals and values [1] and is conceptually associated with locus of control [56]. Cooperativeness is a character trait that refers to the ability to identify with and accept others, and self-transcendence refers to the ability to expand personal boundaries, including experiences of spirituality and feeling oneself as integrated part of the universe as a whole [1,57]. In this study, we assessed subjects’ levels of these character traits using the standardized Korean version of the TCI-Revised [58].

### 4.3. Scan Protocol

All subjects were scanned with [^11^C]DASB using an HRRT-PET scanner (Siemens Medical Imaging Systems, Knoxville, TN, USA). A transmission scan was performed using a ^137^Cs point source prior to [^11^C]DASB injection for attenuation correction. After a bolus injection of 684.0 ± 51.3 MBq [^11^C]DASB with an average specific activity of 47.3 ± 18.3 GBq/μmol (Table 1), emission data were acquired in dynamic mode for 90 min with 22 frames of the following durations: 4 × 30 s, 2 × 60 s, 2 × 90 s, 3 × 150 s, 3 × 210 s, 4 × 300 s, 3 × 600 s, and 1 × 900 s. These emission data were reconstructed using a three-dimensional ordinary Poisson ordered-subset expectation maximization algorithm based on the symmetry and single-instruction multiple-data-based projection and back-projection [59]. The reconstructed PET frames had a voxel size of 1.22 × 1.22 × 1.22 mm^3^ and a matrix size of 256 × 256 × 207. These PET frames were corrected for attenuation, detector dead-time, random and scatter coincidences, decay, and detector normalization.

MRI scans were performed using a 3 Tesla MRI scanner (Magnetom Vida; Siemens Healthcare, Erlangen, Germany) equipped with a 20-channel receiving head/neck coil. A three-dimensional T1-weighted magnetization-prepared rapid gradient echo sequence was used for structural MRI data and the scan parameters were set as follows: repetition time = 1800 ms, echo time = 2.61 ms, inversion time = 900 ms, flip angle = 10°, voxel size = 0.5 × 0.5 × 1.0 mm^3^, matrix size = 512 × 416, and number of slices = 176. The subjects’ heads were fixed as comfortably as possible with sponges to minimize their head movement during the PET and MRI scans.

### 4.4. Image Analysis

Image preprocessing of [^11^C]DASB PET was conducted using Statistical Parametric Mapping 12 (SPM12; The Wellcome Centre for Human Neuroimaging, London, UK; www.fil.ion.ucl.ac.uk (accessed on 1 October 2014)). For each subject, realignment was performed to correct for motion within all reconstructed PET frames, and the structural MRI image was coregistered to an averaged PET image derived from the realignment process. The coregistered structural MRI images and the corresponding PET frames were spatially normalized using the Montreal Neurological Institute template. Based on parameter estimation by kinetic modeling implemented in PMOD software v4.2 (PMOD Technologies Ltd., Zürich, Switzerland), [^11^C]DASB BP_ND_ was obtained using SRTM2 [45] with cerebellar gray matter as the reference region, as suggested in previous studies on this tracer [49,50,51,52,53]. In the [^11^C]DASB BP_ND_ estimation step, regional TACs were extracted from spatially normalized PET frames by averaging all voxels within 52 a priori ROIs associated with the serotonergic system based on a previous PET study [60]. These ROIs included the SFG, middle frontal gyrus (MFG), medial part of the SFG, orbital part of the SFG, medial orbital part of the SFG, orbital part of the MFG, orbital part of the inferior frontal gyrus, OLF, STG, MTG, ITG, SPG, IPG, MOG, anterior cingulate and paracingulate gyri, median cingulate and paracingulate gyri, PCG, HIP, insula, AMYG, caudate nucleus, putamen, nucleus accumbens, globus pallidus, thalamus, dorsal raphe nucleus, and rapheM. Of these, 50 ROIs were predefined by AAL 3 [61], excluding the orbital part of the SFG and the orbital part of the MFG predefined by AAL [62]. Left and right brain regions were evaluated separately to account for laterality in healthy subjects [63].

### 4.5. Statistical Analysis

The ROI-based statistical analyses were conducted using the Statistical Package for the Social Sciences (SPSS) v28.0 (IBM Corp., Armonk, NY, USA). Pearson’s correlation analysis was used to examine the relationships between three character traits, and the equality of variances of [^11^C]DASB BP_ND_ values in each ROI was assessed using Levene’s test. For these analyses, a two-tailed *p* < 0.05 was considered statistically significant.

Several studies have demonstrated that age affects 5-HTT density [64] and character traits [1] and that sex affects 5-HTT binding [65]. Our sample also showed significant sex-dependent differences in the raw score of self-directedness (*p* = 0.024) (Appendix A) and in [^11^C]DASB BP_ND_ values in some brain regions, such as the orbital part of the left SFG (*p* = 0.049), right MTG (*p* = 0.001), right ITG (*p* < 0.001), and bilateral HIP (right: *p* = 0.014, left: *p* = 0.027) (Appendix A). Therefore, to explore the relationship between each character trait and [^11^C]DASB BP_ND_ in brain regions, we performed ROI-based partial correlation analyses with age and sex as covariates, and *p* values less than 0.05 were considered significant. Considering the exploratory perspective of ROI-based partial correlation analysis, if no significant correlation was found at the Bonferroni-corrected *p* < 0.00096 (0.05/52), we further confirmed significant results at a threshold of two-tailed *p* < 0.05. 

To complement the ROI-based results on the relationship between each character trait and [^11^C]DASB BP_ND_ in specific brain regions, we also performed a voxel-based analysis with the same data using SPM12. In this analysis, we set a false discovery rate (FDR)-corrected *p* < 0.05 as the significance threshold for multiple correlations. However, if no significant correlation was found at the FDR-corrected threshold, we further investigated significant results at uncorrected *p* < 0.001 with an extent threshold of 10 voxels, which has been considered statistically significant in previous PET studies [22,66].

## 5. Conclusions

Our findings raise the possibility that there may be significant in vivo correlations between the three character traits and 5-HTT availability in specific brain regions in healthy subjects. In particular, self-directedness was significantly positively associated with 5-HTT availability, suggesting that a goal-oriented, self-confident, and resourceful character may be related to high serotonergic neurotransmission.

## Figures and Tables

**Figure 1 pharmaceuticals-16-00759-f001:**
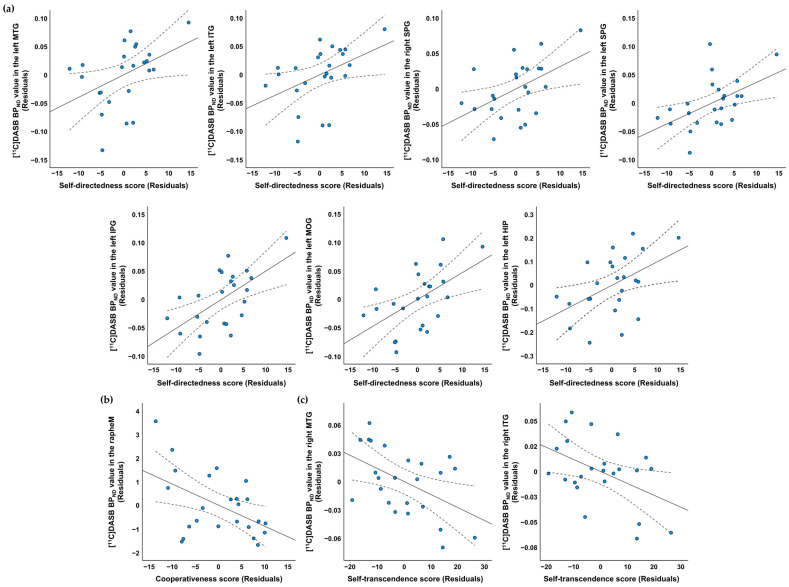
Scatter plots showing the partial correlations between the raw scores of the character scales and [^11^C]DASB BP_ND_ in specific brain regions, controlling for age and sex. (**a**) Self-directedness had significant positive correlations with [^11^C]DASB BP_ND_ in the left MTG (*r* = 0.428, *p* = 0.047), left ITG (*r* = 0.433, *p* = 0.044), bilateral SPG (right: *r* = 0.488, *p* = 0.021, left: *r* = 0.508, *p* = 0.016), left IPG (*r* = 0.593, *p* = 0.004), left MOG (*r* = 0.542, *p* = 0.009), and left HIP (*r* = 0.477, *p* = 0.025). (**b**) Cooperativeness was significantly negatively correlated with [^11^C]DASB BP_ND_ in the rapheM (*r* = −0.497, *p* = 0.019) and (**c**) self-transcendence was significantly negatively correlated with [^11^C]DASB BP_ND_ in the right MTG (*r* = −0.450, *p* = 0.035) and right ITG (*r* = −0.446, *p* = 0.037). The blue dots indicate ordered pairs of the unstandardized residuals obtained from two separate linear regressions of the raw scores of character trait scales and [^11^C]DASB BP_ND_ in specific brain region with respect to age and sex. The solid and dotted lines represent the regression lines and 95% confidence intervals, respectively. BP_ND_, binding potential; MTG, middle temporal gyrus; ITG, inferior temporal gyrus; SPG, superior parietal gyrus; IPG, inferior parietal gyrus; MOG, middle occipital gyrus; HIP, hippocampus; rapheM, median raphe nucleus.

**Figure 2 pharmaceuticals-16-00759-f002:**
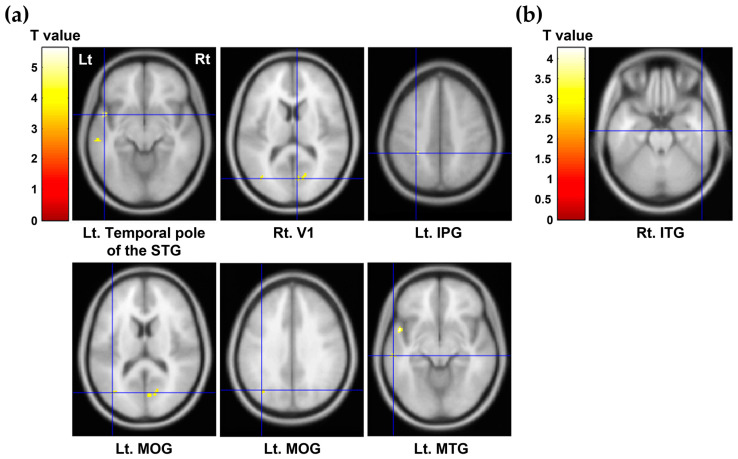
Results of voxel-based partial correlation analysis between the raw scores of the character traits and [^11^C]DASB BP_ND_ in the whole brain (uncorrected *p* < 0.001). (**a**) Self-directedness was significantly positively correlated with [^11^C]DASB BP_ND_ in the temporal pole of the left STG (peak MNI coordinate: −50 6 −10, cluster size = 26 voxels, *p* < 0.0001), right V1 (peak MNI coordinate = 6 −74 12, cluster size = 67 voxels, *p* = 0.0001), left IPG (peak MNI coordinate = −30 −42 46, cluster size = 24 voxels, *p* = 0.0001), left MOG (bottom left of the figure: peak MNI coordinate = −40 −72 14, cluster size = 16 voxels, *p* = 0.0002, bottom center of the figure: peak MNI coordinate = −38 −68 34, cluster size = 16 voxels, *p* = 0.0002), and left MTG (peak MNI coordinate = −58 −26 −10, cluster size = 15 voxels, *p* = 0.0002). (**b**) Self-transcendence had a significant negative correlation with [^11^C]DASB BP_ND_ in the right ITG (peak MNI coordinate = 52 −14 −24, cluster size = 15 voxels; *p* = 0.0002). BP_ND_, binding potential; STG, superior temporal gyrus; MNI, Montreal Neurological Institute; V1, calcarine fissure and surrounding cortex; IPG, inferior parietal gyrus; MOG, middle occipital gyrus; MTG, middle temporal gyrus; ITG, inferior temporal gyrus; Lt, left; Rt, right.

**Table 1 pharmaceuticals-16-00759-t001:** Demographic characteristics and PET scan information of healthy subjects.

Variables	Mean ± SD/Number (%)
Demographic characteristics	
Age (years)	29.9 ± 7.9
Sex	
Male	12 (50%)
Female	12 (50%)
Education (years)	15.0 ± 1.6
Marital status	
Single	16 (66.7%)
Married	8 (33.3%)
Raw scores of TCI character scales	
Self-directedness	54.7 ± 7.7
Cooperativeness	59.5 ± 7.9
Self-transcendence	22.7 ± 13.3
[^11^C]DASB PET scan information	
Injected dose (MBq)	684.0 ± 51.3
Specific activity (GBq/μmol)	47.3 ± 18.3

PET, positron emission tomography; SD, standard deviation; TCI, Temperament and Character Inventory.

**Table 2 pharmaceuticals-16-00759-t002:** Mean [^11^C]DASB BP_ND_ values and their correlation coefficients with raw scores of character scales in ROIs.

Regions of Interest	Mean (SD)	Correlation Coefficients ^†^ (*p* Value)		
		Self-Directedness	Cooperativeness	Self-Transcendence
Rt. middle temporal gyrus	0.29 (0.04)	−0.092 (0.683)	0.095 (0.674)	−0.450 (0.035) *
Lt. middle temporal gyrus	0.40 (0.06)	0.428 (0.047) *	−0.004 (0.987)	−0.045 (0.842)
Rt. inferior temporal gyrus	0.33 (0.06)	−0.042 (0.854)	−0.112 (0.619)	−0.446 (0.037) *
Lt. inferior temporal gyrus	0.41 (0.05)	0.433 (0.044) *	−0.045 (0.841)	−0.170 (0.449)
Rt. superior parietal gyrus	0.23 (0.04)	0.488 (0.021) *	0.022 (0.924)	−0.122 (0.590)
Lt. superior parietal gyrus	0.23 (0.04)	0.508 (0.016) *	0.066 (0.771)	0.025 (0.913)
Lt. inferior parietal gyrus	0.30 (0.05)	0.593 (0.004) **	0.231 (0.301)	−0.026 (0.907)
Lt. middle occipital gyrus	0.32 (0.05)	0.542 (0.009) **	−0.123 (0.584)	−0.053 (0.815)
Lt. hippocampus	0.96 (0.14)	0.477 (0.025) *	0.232 (0.299)	−0.150 (0.505)
Median raphe nucleus	2.06 (1.34)	−0.245 (0.272)	−0.497 (0.019) *	−0.155 (0.491)

^†^ Correlation coefficients were obtained using ROI-based partial correlation analysis with age and sex as covariates. Asterisks indicate statistical significance at *p* < 0.05 * and *p* < 0.01 **. BP_ND_, binding potential; ROI, region of interest; SD, standard deviation; Rt, right; Lt, left.

## Data Availability

Data sharing not applicable.

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
