# Peer review of "The Relationship between Character Traits and In Vivo Cerebral Serotonin Transporter Availability in Healthy Subjects: A High-Resolution PET Study with C-11 DASB"

_pharmaceuticals, 2023, doi:10.3390/ph16050759_

Round 1

Reviewer 1 Report

-although the sample size is limited, can you perform a supplementary analysis stratified by sex rather then entered as a covariate?

-Can you clarify the "NA" entries in table 2? Is this because of the lack of signal in those regions and therefore unable to perform the analysis?

-can you discuss your findings in the serotonin system compared to other neurotransmitter systems? Have other studies been completed with these traits or personality traits and other neurotransmitters such as dopamine.

Author Response

Comment 1:

Although the sample size is limited, can you perform a supplementary analysis stratified by sex rather then entered as a covariate?

Response to comment 1:

As suggested, we have conducted a supplementary analysis stratified by sex and presented the results as Supplementary Table S4 in the Supplementary Materials. We have also added the following paragraph to the manuscript.

“We further investigated the relationship between character traits and 5-HTT availability in ROIs for males and females separately using ROI-based correlation analysis with age as a covariate. As shown in Supplementary Tables S4, males had significant negative correlations between self-transcendence and 5-HTT availability in the right STG, right MTG, and right ITG. Females exhibited significant positive correlations between self-directedness and 5-HTT availability in the medial orbital part of left superior frontal gyrus (SFG), left olfactory cortex (OLF), left SPG, left IPG, left HIP, and right AMYG. Further studies on larger cohorts are required owing to limited sample size (12 males and 12 females) in additional supplementary analyses stratified by sex.” (Page 7/3. Discussion/lines 217-225)

Comment 2:

Can you clarify the "NA" entries in table 2? Is this because of the lack of signal in those regions and therefore unable to perform the analysis?

Response to comment 2:

In Table 2 of the initial submission, we denoted the non-significant correlation between the [11C]DASB BPND and the scores of character scales as NA. As suggested, to present our results more clearly, we have changed NA entries to correlation coefficients and p values. (Page 3/Table 2/lines 101-104)

Comment 3:

Can you discuss your findings in the serotonin system compared to other neurotransmitter systems? Have other studies been completed with these traits or personality traits and other neurotransmitters such as dopamine.

Response to comment 3:

Most human brain molecular PET imaging studies on neurotransmitters such as dopamine [1–5], 5-HT [6–12], glutamate [13], and acetylcholine [14] have examined associations with temperament traits not character traits.

Until recently, only one PET study using [18F]Fallypride investigated associations between self-transcendence (i.e., one of character traits) and dopaminergic system, and reported a significant positive association between self-transcendence trait and dopamine D2/3 receptor availability in vivo in the left insula in healthy subjects [15].

Because, to the best of our knowledge, only one study has examined associations between character traits and other neurotransmitters, it is not adequate to discuss our findings in the serotonin system compared to that in other neurotransmitter systems. However, a previous [18F]Fallypride PET study by Kim et al. [15] and our study suggest that the self-transcendence trait may have differential associations with different neurotransmitters in specific brain regions. Further molecular PET imaging studies are needed to gain more insights into the relationships between character traits and various neurotransmitters.

[References]

  1. Suhara, T.; Yasuno, F.; Sudo, Y.; Yamamoto, M.; Inoue, M.; Okubo, Y.; Suzuki, K. Dopamine D2 receptors in the insular cortex and the personality trait of novelty seeking. Neuroimage 2001, 13 (5), 891-895. DOI: 10.1006/nimg.2001.0761.
  2. Yasuno, F.; Suhara, T.; Sudo, Y.; Yamamoto, M.; Inoue, M.; Okubo, Y.; Suzuki, K. Relation among dopamine D-2 receptor binding, obesity and personality in normal human subjects. Neurosci Lett 2001, 300 (1), 59-61. DOI: 10.1016/S0304-3940(01)01552-X.
  3. Kim, J. H.; Son, Y. D.; Kim, H. K.; Lee, S. Y.; Cho, S. E.; Kim, Y. B.; Cho, Z. H. Association of harm avoidance with dopamine D(2/3) receptor availability in striatal subdivisions: A high resolution PET study. Biol Psychol 2011, 87 (1), 164-167. DOI: 10.1016/j.biopsycho.2011.02.011.
  4. Caravaggio, F.; Chung, J. K.; Gerretsen, P.; Fervaha, G.; Nakajima, S.; Plitman, E.; Iwata, Y.; Wilson, A.; Graff-Guerrero, A. Exploring the relationship between social attachment and dopamine D-2/3 receptor availability in the brains of healthy humans using [C-11]-(+)-PHNO. Soc Neurosci-Uk 2017, 12 (2), 163-173. DOI: 10.1080/17470919.2016.1152997.
  5. Smith, C. T.; San Juan, M. D.; Dang, L. C.; Katz, D. T.; Perkins, S. F.; Burgess, L. L.; Cowan, R. L.; Manning, H. C.; Nickels, M. L.; Claassen, D. O.; et al. Ventral striatal dopamine transporter availability is associated with lower trait motor impulsivity in healthy adults. Transl Psychiat 2018, 8. DOI: 10.1038/s41398-018-0328-y.
  6. Borg, J.; Andree, B.; Soderstrom, H.; Farde, L. The serotonin system and spiritual experiences. Am J Psychiat 2003, 160 (11), 1965-1969. DOI: 10.1176/appi.ajp.160.11.1965.
  7. Bailer, U. F.; Price, J. C.; Meltzer, C. C.; Mathis, C. A.; Frank, G. K.; Weissfeld, L.; McConaha, C. W.; Henry, S. E.; Brooks-Achenbach, S.; Barbarich, N. C.; et al. Altered 5-HT2A receptor binding after recovery from bulimia-type anorexia nervosa: Relationships to harm avoidance and drive for thinness. Neuropsychopharmacol 2004, 29 (6), 1143-1155. DOI: 10.1038/sj.npp.1300430.
  8. Reimold, M.; Batra, A.; Knobel, A.; Smolka, M. N.; Zimmer, A.; Mann, K.; Solbach, C.; Reischl, G.; Schwarzler, F.; Grunder, G.; et al. Anxiety is associated with reduced central serotonin transporter availability in unmedicated patients with unipolar major depression: a [C-11]DASB PET study. Mol Psychiatr 2008, 13 (6), 606-613. DOI: 10.1038/sj.mp.4002149.
  9. Gerretsen, P.; Graff-Guerrero, A.; Menon, M.; Pollock, B. G.; Kapur, S.; Vasdev, N.; Houle, S.; Mamo, D. Is desire for social relationships mediated by the serotonergic system in the prefrontal cortex? An [18F]setoperone PET study. Soc Neurosci-Uk 2010, 5 (4), 375-383. DOI: 1080/17470911003589309.
  10. Soloff, P. H.; Price, J. C.; Mason, N. S.; Becker, C.; Meltzer, C. C. Gender, personality, and serotonin-2A receptor binding in healthy subjects. Psychiat Res-Neuroim 2010, 181 (1), 77-84. DOI: 10.1016/j.pscychresns.2009.08.007.
  11. Tuominen, L.; Salo, J.; Hirvonen, J.; Nagren, K.; Laine, P.; Melartin, T.; Isometsa, E.; Viikari, J.; Cloninger, C. R.; Raitakari, O.; et al. Temperament, character and serotonin activity in the human brain: a positron emission tomography study based on a general population cohort. Psychol Med 2013, 43 (4), 881-894. DOI: 10.1017/S003329171200164x.
  12. Soloff, P. H.; Chiappetta, L.; Mason, N. S.; Becker, C.; Price, J. C. Effects of serotonin-2A receptor binding and gender on personality traits and suicidal behavior in borderline personality disorder. Psychiat Res-Neuroim 2014, 222 (3), 140-148. DOI: 10.1016/j.pscychresns.2014.03.008.
  13. Leurquin-Sterk, G.; Van den Stock, J.; Crunelle, C. L.; de Laat, B.; Weerasekera, A.; Himmelreich, U.; Bormans, G.; Van Laere, K. Positive Association Between Limbic Metabotropic Glutamate Receptor 5 Availability and Novelty-Seeking Temperament in Humans: An F-18-FPEB PET Study. J Nucl Med 2016, 57 (11), 1746-1752. DOI: 10.2967/jnumed.116.176032.
  14. Storage, S.; Mandelkern, M. A.; Phuong, J.; Kozman, M.; Neary, M. K.; Brody, A. L. A positive relationship between harm avoidance and brain nicotinic acetylcholine receptor availability. Psychiat Res-Neuroim 2013, 214 (3), 415-421. DOI: 10.1016/j.pscychresns.2013.07.010.
  15. Kim, J. H.; Choe, Y. S.; Cumming, P.; Son, Y. D.; Kim, H. K.; Joo, Y. H.; Kim, J. H. Relationship of self-transcendence traits with in vivo dopamine D2/3 receptor availability and functional connectivity: An [F-18]fallypride PET and fMRI study. Synapse 2019, 73 (11). DOI: 10.1002/syn.22121.

Reviewer 2 Report

This study examined the association between serotonin transporter availability (determined as binding potential in PET scans), and character traits defined by Cloninger’s Temperament and Character Inventory (TCI). Binding potential (BP) was measured by a tissue reference method, a technique not without its critics and inferior to arterial blood sampling. Whether the current results are affected by the differences that might ensue as a result of technique is not clear, although the authors argue to the contrary (see lines 232 ff).  

While one can appreciate that Cloninger suggested a relationship between some domains of the TCI and certain neurotransmitter activity this has not been demonstrated empirically, although such domains have been associated with genetic variants of some aspects of neurotransmitter activity (e.g., novelty seeking and DAT1). In essence this study sought to examine these putative relationships in detail. Given the specificity of Cloninger’s hypothesis, namely that serotonergic activity is associated with harm avoidance, it is surprising that the authors have not addressed this notion specifically and rather concentrated on other domains of the TCI. The logic for the examining the domains of “self-directedness”, “self-transcendence” and “cooperativeness” seems somewhat at odds with examining the serotonergic system specifically other than the tenuous connection between these domains, psychological well-being and serotonin. Given that the subjects of the study were healthy individuals the range of “psychological well-being” is likely to be rather small and any implications drawn from the study tenuous.

Although the number of subjects for an imaging study of this kind is quite large, in terms the power of the study, it is probably quite small. Multiple correlations have been performed without it seems any correction for such. In any event, the observed correlations are quite small and explain relatively little of the variance. Examination of the scattergrams suggest a wide dispersion of results which begs the question of whether a large sample size may not have diminished the “strength” of the associations found.  There seem to be a number of outliers in these scattergrams which perhaps weight the associations?

Some specific comments:

1. Line 160: suggest a causal relationship between 5HTT availability and self-directedness. Is this “cause and effect” conclusion too strong or is it more simply stated as an association?   

2. Line 164ff: It is not clear how the present results are consistent with life satisfaction and positive emotions when these were not measured in the study.

3. Line 168ff: Statement seems contradictory. On the one hand it is sated that there is a significant correlation in the right calcarine fissure and surrounding cortex, one of the brain regions excluded from the ROIs in this study. If the region is excluded how can there be a correlation?

4. Lines 181-185: The meaning of this sentence here is obscure. It seems to be a circular argument. I would suggest breaking the sentence up to make the meaning clearer for the reader.

5. Line 352: The authors introduce the concept of ‘locus of control’ and associate this with 5HTT availability and self-directedness without either any theoretical link being discussed or any measurement of locus of control in their subjects.

The English usage is good for the most part. There are one or two sentences where the meaning is not entirely clear to me. 

Author Response

Comment 1:

This study examined the association between serotonin transporter availability (determined as binding potential in PET scans), and character traits defined by Cloninger’s Temperament and Character Inventory (TCI). Binding potential (BP) was measured by a tissue reference method, a technique not without its critics and inferior to arterial blood sampling. Whether the current results are affected by the differences that might ensue as a result of technique is not clear, although the authors argue to the contrary (see lines 232).

Response to comment 1:

We acknowledge that metabolite-corrected arterial input-based quantification is the ideal method. However, quantification of [11C]DASB BPND using arterial blood sampling is invasive as it requires radial artery cannulation, and consequent discomforts might be a confounding factor. The recruitment of participants for studies involving arterial blood sampling is often limited. Inaccuracies in the measurement of plasma metabolite fractions can also be a source of bias in endpoint estimation [1]. Therefore, we quantified [11C]DASB BPND using the reference tissue model, which has been used as an established method in [11C]DASB PET [2–6]. Further molecular PET studies employing arterial input-based quantification methods are required to confirm our findings on the relationships between character traits and serotonin transporter availability.

We have added the following paragraph to the manuscript.

“Metabolite-corrected arterial input-based kinetic modeling methods are ideal for quantification of [11C]DASB BPND; however, these methods are invasive as they require radial artery cannulation, and the resulting discomforts might be a confounding factor and would also have limited recruitment of participants. Inaccuracies in determining the arterial input function can also be a source of bias in endpoint estimation [1].” (Page 7/3. Discussion/lines 233-238) “Further molecular PET studies using arterial input-based kinetic modeling methods are required to confirm our results on the relationships between character traits and 5-HTT availability.” (Page 7/3. Discussion/lines 251-253)

[References]

  1. Slifstein, M.; Laruelle, M. Models and methods for derivation of in vivo neuroreceptor parameters with PET and SPECT reversible radiotracers. Nucl Med Biol 2001, 28 (5), 595-608. DOI: 10.1016/S0969-8051(01)00214-1.
  2. Ichise, M.; Liow, J. S.; Lu, J. Q.; Takano, T.; Model, K.; Toyama, H.; Suhara, T.; Suzuki, T.; Innis, R. B.; Carson, T. E. Linearized reference tissue parametric Imaging methods: Application to [C-11]DASB positron emission tomography studies of the serotonin transporter in human brain. J Cerebr Blood F Met 2003, 23 (9), 1096-1112. DOI: 10.1097/01.Wcb.0000085441.37552.Ca.
  3. Praschak-Rieder, N.; Wilson, A. A.; Hussey, D.; Carella, A.; Wei, C.; Ginovart, N.; Schwarz, M. J.; Zach, J.; Houle, S.; Meyer, J. H. Effects of tryptophan depletion on the serotonin transporter in healthy humans. Biol Psychiat 2005, 58 (10), 825-830. DOI: 10.1016/j.biopsych.2005.04.038.
  4. Tyrer, A. E.; Levitan, R. D.; Houle, S.; Wilson, A. A.; Nobrega, J. N.; Meyer, J. H. Increased Seasonal Variation in Serotonin Transporter Binding in Seasonal Affective Disorder. Neuropsychopharmacol 2016, 41 (10), 2447-2454. DOI: 10.1038/npp.2016.54.
  5. Smit, M.; Vallez Garcia, D.; de Jong, B. M.; Zoons, E.; Booij, J.; Dierckx, R. A.; Willemsen, A. T.; de Vries, E. F.; Bartels, A. L.; Tijssen, M. A. Relationships between Serotonin Transporter Binding in the Raphe Nuclei, Basal Ganglia, and Hippocampus with Clinical Symptoms in Cervical Dystonia: A [C-11]DASB Positron Emission Tomography Study. Front Neurol 2018, 9. DOI: 10.3389/fneur.2018.00088.
  6. Timmers, E. R.; Peretti, D. E.; Smit, M.; de Jong, B. M.; Dierckx, R.; Kuiper, A.; de Koning, T. J.; Vallez Garcia, D.; Tijssen, M. A. J. Serotonergic system in vivo with [(11)C]DASB PET scans in GTP-cyclohydrolase deficient dopa-responsive dystonia patients. Sci Rep 2022, 12 (1), 6292. DOI: 10.1038/s41598-022-10067-5.

Comment 2:

While one can appreciate that Cloninger suggested a relationship between some domains of the TCI and certain neurotransmitter activity this has not been demonstrated empirically, although such domains have been associated with genetic variants of some aspects of neurotransmitter activity (e.g., novelty seeking and DAT1). In essence this study sought to examine these putative relationships in detail. Given the specificity of Cloninger’s hypothesis, namely that serotonergic activity is associated with harm avoidance, it is surprising that the authors have not addressed this notion specifically and rather concentrated on other domains of the TCI.

Response to comment 2:

As we mentioned in the initial submission, several genetic association studies have suggested associations between human character traits and the serotonergic system [1–4], but these results have not yet been validated in in vivo molecular PET imaging studies. Therefore, our exploratory study examined the association between human character traits and 5-HTT availability in vivo using [11C]DASB PET.

Based on your comments, we have performed an additional supplementary ROI-based analysis with age and sex as covariates to identify the association between the harm avoidance temperament and the 5-HTT availability in vivo and presented the methods and results in the Supplementary Materials. We have also added the following paragraph to the manuscript.

“Considering that the harm avoidance temperament has been suggested to be associated with serotonergic neurotransmission [5–9], we performed an additional supplementary ROI-based analysis with age and sex as covariates. We found a significant negative association between the harm avoidance temperament and the 5-HTT availability in the left posterior cingulate gyrus (p = 0.028) (Supplementary Figure S1), suggesting the association between the harm avoidance temperament and the pre-synaptic marker of serotonergic neurotransmission.” (Page 7/3. Discussion/lines 226-232)

[References]

  1. Comings, D. E.; Gade-Andavolu, R.; Gonzalez, N.; Wu, S.; Muhleman, D.; Blake, H.; Mann, M. B.; Dietz, G.; Saucier, G.; MacMurray, J. P. A multivariate analysis of 59 candidate genes in personality traits: the temperament and character inventory. Clin Genet 2000, 58 (5), 375-385. DOI: 10.1034/j.1399-0004.2000.580508.x.
  2. Gonda, X.; Fountoulakis, K. N.; Juhasz, G.; Rihmer, Z.; Lazary, J.; Laszik, A.; Akiskal, H. S.; Bagdy, G. Association of the s allele of the 5-HTTLPR with neuroticism-related traits and temperaments in a psychiatrically healthy population. Eur Arch Psy Clin N 2009, 259 (2), 106-113. DOI: 10.1007/s00406-008-0842-7.
  3. Alfimova, M. V.; Monakhov, M. V.; Golimbet, V. E.; Korovaitseva, G. I.; Lyashenko, G. L. Analysis of Associations between 5-HTT, 5-HTR2A, and GABRA6 Gene Polymorphisms and Health-Associated Personality Traits. B Exp Biol Med+ 2010, 149 (4), 434-436. DOI: 10.1007/s10517-010-0964-6.
  4. Calati, R.; Signorelli, M. S.; Gressier, F.; Bianchini, O.; Porcelli, S.; Comings, D. E.; De Girolamo, G.; Aguglia, E.; MacMurray, J.; Serretti, A. Modulation of a number of genes on personality traits in a sample of healthy subjects. Neurosci Lett 2014, 566, 320-325. DOI: 10.1016/j.neulet.2014.02.001.
  5. Cloninger, C. R.; Svrakic, D. M.; Przybeck, T. R. A Psychobiological Model of Temperament and Character. Arch Gen Psychiat 1993, 50 (12), 975-990. DOI: 10.1001/archpsyc.1993.01820240059008.
  6. Comings, D. E.; Gade-Andavolu, R.; Gonzalez, N.; Wu, S.; Muhleman, D.; Blake, H.; Mann, M. B.; Dietz, G.; Saucier, G.; MacMurray, J. P. A multivariate analysis of 59 candidate genes in personality traits: the temperament and character inventory. Clin Genet 2000, 58 (5), 375-385. DOI: 10.1034/j.1399-0004.2000.580508.x.
  7. Mazzanti, C. M.; Lappalainen, J.; Long, J. C.; Bengel, D.; Naukkarinen, H.; Eggert, M.; Virkkunen, M.; Linnoila, M.; Goldman, D. Role of the serotonin transporter promoter polymorphism in anxiety-related traits. Arch Gen Psychiat 1998, 55 (10), 936-940. DOI: 10.1001/archpsyc.55.10.936.
  8. Wiesbeck, G. A.; Weijers, H. G.; Wodarz, N.; Keller, H. K.; Michel, T. M.; Herrmann, M. J.; Boening, J. Serotonin transporter gene polymorphism and personality traits in primary alcohol dependence. World J Biol Psychia 2004, 5 (1), 45-48. DOI: 10.1080/15622970410029907.
  9. Schneider-Matyka, D.; Jurczak, A.; Szkup, M.; Samochowiec, A.; Grzywacz, A.; Wieder-Huszla, S.; Grochans, E. The influence of the serotonergic system on the personality and quality of life of postmenopausal women. Clin Interv Aging 2017, 12, 963-970. DOI: 10.2147/Cia.S133712.

Comment 3:

The logic for the examining the domains of “self-directedness”, “self-transcendence” and “cooperativeness” seems somewhat at odds with examining the serotonergic system specifically other than the tenuous connection between these domains, psychological well-being and serotonin. Given that the subjects of the study were healthy individuals the range of “psychological well-being” is likely to be rather small and any implications drawn from the study tenuous.

Response to comment 3:

We acknowledge the comment that the range of psychological well-being in healthy subjects may be rather small; therefore, the implications drawn may be tenuous.

The objective of our exploratory study was to investigate the associations between human character traits and 5-HTT availability in vivo using [11C]DASB PET. As we mentioned in the initial submission, this investigation was based on the genetic association studies that have previously demonstrated the relationships between character traits and serotonergic system in humans [1–4]. In line with this rationale of our study, we found significant associations between the character traits and the 5-HTT availability measured with [11C]DASB PET. 

Considering your comment, we have deleted the following sentences regarding the implications of our findings for the relationships with well-being and affective states, which were not directly measured in our study.

“Two studies of human character traits found that higher self-directedness was consistently associated with higher affective and non-affective well-being, including life satisfaction, social support, subjective health, positive affect, and negative affect, regardless of interactions with other traits, while lower self-directedness (and sometimes lower cooperativeness) was associated with higher negative affect [3,41]. Based on these previous reports, we expect that 5-HTT availability in healthy subjects would be positively correlated with both self-directedness and cooperativeness; however, the present study reported that 5-HTT availability in healthy subjects was positively correlated with self-directedness and negatively correlated with cooperativeness. Future studies on larger cohorts are needed to validate our molecular neuroimaging results.”

[References]

  1. Comings, D. E.; Gade-Andavolu, R.; Gonzalez, N.; Wu, S.; Muhleman, D.; Blake, H.; Mann, M. B.; Dietz, G.; Saucier, G.; MacMurray, J. P. A multivariate analysis of 59 candidate genes in personality traits: the temperament and character inventory. Clin Genet 2000, 58 (5), 375-385. DOI: 10.1034/j.1399-0004.2000.580508.x.
  2. Gonda, X.; Fountoulakis, K. N.; Juhasz, G.; Rihmer, Z.; Lazary, J.; Laszik, A.; Akiskal, H. S.; Bagdy, G. Association of the s allele of the 5-HTTLPR with neuroticism-related traits and temperaments in a psychiatrically healthy population. Eur Arch Psy Clin N 2009, 259 (2), 106-113. DOI: 10.1007/s00406-008-0842-7.
  3. Alfimova, M. V.; Monakhov, M. V.; Golimbet, V. E.; Korovaitseva, G. I.; Lyashenko, G. L. Analysis of Associations between 5-HTT, 5-HTR2A, and GABRA6 Gene Polymorphisms and Health-Associated Personality Traits. B Exp Biol Med+ 2010, 149 (4), 434-436. DOI: 10.1007/s10517-010-0964-6.
  4. Calati, R.; Signorelli, M. S.; Gressier, F.; Bianchini, O.; Porcelli, S.; Comings, D. E.; De Girolamo, G.; Aguglia, E.; MacMurray, J.; Serretti, A. Modulation of a number of genes on personality traits in a sample of healthy subjects. Neurosci Lett 2014, 566, 320-325. DOI: 10.1016/j.neulet.2014.02.001.

Comment 4:

Although the number of subjects for an imaging study of this kind is quite large, in terms the power of the study, it is probably quite small. Multiple correlations have been performed without it seems any correction for such. In any event, the observed correlations are quite small and explain relatively little of the variance. Examination of the scattergrams suggest a wide dispersion of results which begs the question of whether a large sample size may not have diminished the “strength” of the associations found. There seem to be a number of outliers in these scattergrams which perhaps weight the associations?

Response to comment 4:

We acknowledge the comment that the observed correlations are limited because of the relatively wide dispersion of scatters in the scatter plot. Considering your comment, we have incorporated the following paragraph into the manuscript as a limitation of our study.

“As illustrated in Figure 1, the scores of character traits and [11C]DASB BPND values exhibited substantial variances. The wide dispersion of data points may have resulted in relatively low statistical power in examining the observed correlations. Consequently, our study may have been limited in detecting relatively weak associations between character traits and 5-HTT availability as assessed using the ROI-based correlation analysis.” (Page 7/3. Discussion/lines 254-259)

Specific comment 1:

Line 160: suggest a causal relationship between 5HTT availability and self-directedness. Is this “cause and effect” conclusion too strong or is it more simply stated as an association?  

Response to specific comment 1:

Based on your comments, we have revised the sentence as follows to describe the relationship as an association not a causal effect.

“The positive correlation between self-directedness and 5-HTT availability in vivo in these brain regions may be associated with genetic factors affecting 5-HTT or with secondary changes in 5-HTT density due to changes in endogenous extracellular serotonin levels.” (Page 6/3. Discussion/lines 160-163)

Specific comment 2:

Line 164: It is not clear how the present results are consistent with life satisfaction and positive emotions when these were not measured in the study.

Response to specific comment 2:

Based on your comments, since we did not directly measure the levels of life satisfaction and positive emotions, we have deleted the sentence in the revised manuscript.

Specific comment 3:

Line 168: Statement seems contradictory. On the one hand it is sated that there is a significant correlation in the right calcarine fissure and surrounding cortex, one of the brain regions excluded from the ROIs in this study. If the region is excluded how can there be a correlation?

Response to specific comment 3:

We intended to describe that the region (the right calcarine fissure and surrounding cortex, labeled right V1 in the revised manuscript) was not included in the 52 a priori regions of interest (ROIs). We have revised the sentence as follows to avoid confusion.

“As shown in Figure 2 and Supplementary Table S3, additional automated anatomical labeling (AAL) template region- and voxel-based partial correlation analyses also revealed a significant positive correlation between self-directedness and 5-HTT availability in the right V1, one of the brain regions excluded from the 52 a priori ROIs in this study.” (Page 6/3. Discussion/lines 165-168)

Specific comment 4:

Lines 181-185: The meaning of this sentence here is obscure. It seems to be a circular argument. I would suggest breaking the sentence up to make the meaning clearer for the reader.

Response to specific comment 4:

Based on your comments, we have revised the sentence as follows to clarify the meaning.

“A recent study by Kim et al. demonstrated that self-directedness was predicted by low sensory registration and high sensory seeking [27]. These results suggest that specific sensory processing patterns may influence the trait of self-directedness [27].” (Page 6/3. Discussion/lines 178-181)

Specific comment 5:

Line 352: The authors introduce the concept of ‘locus of control’ and associate this with 5HTT availability and self-directedness without either any theoretical link being discussed or any measurement of locus of control in their subjects.

Response to specific comment 5:

In the initial submission, we described high self-directedness as a character with the internal locus of control since self-directedness is conceptually and empirically associated with the locus of control [1]. In a psychobiological model of temperament and character proposed by Cloninger, high self-directedness was associated with the internal locus of control, while low self-directedness was related to the external locus of control [1]. However, we did not separately measure the locus of control in our study. Therefore, based on your comments, we have deleted the term “with the internal locus of control” in the revised manuscript to avoid confusion.

[Reference]

  1. Cloninger, C. R.; Svrakic, D. M.; Przybeck, T. R. A Psychobiological Model of Temperament and Character. Arch Gen Psychiat 1993, 50 (12), 975-990. DOI: 10.1001/archpsyc.1993.01820240059008.

Round 2

Reviewer 2 Report

The ms has been improved by the alterations which the authors have made. The limited nature of the associations found have been acknowledged.